# Multiphoton Microscopy for Identifying Collagen Signatures Associated with Biochemical Recurrence in Prostate Cancer Patients

**DOI:** 10.3390/jpm11111061

**Published:** 2021-10-22

**Authors:** Ina P. Pavlova, Sujit S. Nair, Dara Lundon, Stanislaw Sobotka, Reza Roshandel, Patrick-Julien Treacy, Parita Ratnani, Rachel Brody, Jonathan I. Epstein, Gustavo E. Ayala, Natasha Kyprianou, Ashutosh K. Tewari

**Affiliations:** 1Department of Urology, Icahn School of Medicine at Mount Sinai, New York, NY 10029, USA; sujit.nair@mountsinai.org (S.S.N.); Dara.Lundon@mountsinai.org (D.L.); STANISLAW.SOBOTKA@mountsinai.org (S.S.); reza.roshandel@mountsinai.org (R.R.); Parita.Ratnani@mountsinai.org (P.R.); Natasha.Kyprianou@mountsinai.org (N.K.); 2Department of Urology, Pasteur 2 University Hospital of Nice, 06000 Nice, France; pj.treacy@live.fr; 3Department of Pathology, Molecular and Cell Based Medicine, Icahn School of Medicine at Mount Sinai, New York, NY 10029, USA; Rachel.Brody@mountsinai.org; 4Department of Pathology, Urology and Oncology, Johns Hopkins Hospital, Baltimore, MD 21287, USA; jepstein@jhmi.edu; 5Department of Pathology and Laboratory Medicine, University of Texas Health Science Center, Houston, TX 77030, USA; Gustavo.E.Ayala@uth.tmc.edu; 6Department of Oncological Sciences, Tisch Cancer Institute, Icahn School of Medicine at Mount Sinai, New York, NY 10029, USA

**Keywords:** optical microscopy, prostate cancer, reactive stroma, collagen signatures, prognosis

## Abstract

Prostate cancer is a heterogeneous disease that remains dormant for long periods or acts aggressively with poor clinical outcomes. Identifying aggressive prostate tumor behavior using current glandular-focused histopathological criteria is challenging. Recent evidence has implicated the stroma in modulating prostate tumor behavior and in predicting post-surgical outcomes. However, the emergence of stromal signatures has been limited, due in part to the lack of adoption of imaging modalities for stromal-specific profiling. Herein, label-free multiphoton microscopy (MPM), with its ability to image tissue with stromal-specific contrast, is used to identify prostate stromal features associated with aggressive tumor behavior and clinical outcome. MPM was performed on unstained prostatectomy specimens from 59 patients and on biopsy specimens from 17 patients with known post-surgery recurrence status. MPM-identified collagen content, organization, and morphological tumor signatures were extracted for each patient and screened for association with recurrent disease. Compared to tumors from patients whose disease did not recur, tumors from patients with recurrent disease exhibited higher MPM-identified collagen amount and collagen fiber intensity signal and width. Our study shows an association between MPM-identified stromal collagen features of prostate tumors and post-surgical disease recurrence, suggesting their potential for prostate cancer risk assessment.

## 1. Introduction

Prostate cancer (PCa) is the second most diagnosed cancer and the third leading cause of cancer death in men annually [1]. It can remain indolent for long periods or can rapidly progress to lethal disease, accounting for its high annual death rate [1,2]. Early identification of aggressive versus indolent PCa is challenging, in part, because current clinicopathologic criteria and models are limited in predicting cancer behavior and post-surgery clinical outcomes for low- and intermediate-risk patients [3]. Although existing grading criteria focus on glandular tissue features [4,5], growing evidence supports the role of stroma-epithelial crosstalk in tumor initiation and progression [6,7,8,9] and that of the stroma in modulating PCa behavior [3,10]. During PCa development, the stroma becomes reactive [11,12]. A new grading system has been developed that quantifies the amount of activated stroma in tumors using well-described features, including loss of smooth muscle cells, expansion of the extracellular matrix (ECM), and activation of myofibroblasts [13]. Studies using biopsy [14] and radical prostatectomy specimens [15] have shown that a large amount (>50%) of intratumor reactive stroma predicts post-operative recurrence. Moreover, with tumor progression, the ECM becomes stiff and aligned [16], and quantifiers of the orientation and alignment of collagen fibers, the main component of the ECM, have been associated with tumor aggressiveness in PCa [17,18] and with poor prognosis in other human cancers [19,20,21]. Although there is increasing support for including stromal characteristics in clinicopathological models, PCa nodules are currently detected by relatively low-resolution visualization approaches, such as multiparametric magnetic resonance imaging, followed by a biopsy, and then by examination of the hematoxylin and eosin (H&E)-stained tissue. This approach does not facilitate routine detection and quantification of subtle changes in stromal morphology and thus limits the identification of new features that hold promise for differentiating between aggressive and indolent cancer behavior. There is an unmet clinical need to identify such features using imaging modalities that capture both prostate stromal and glandular changes, offer subcellular resolution, ECM-specific contrast, and integrate them into existing clinical imaging protocols.

Optical, nonlinear imaging modalities, such as multiphoton microscopy (MPM), are appropriate for high-resolution interrogation of tumor microanatomy, especially in the context of offering intrinsic, collagen-enhanced contrast alternatives to standard H&E assessment. Due to its capture of intrinsic contrasts, and other advantages such as subcellular, high-contrast optical sectioning with minimal photodamage and photobleaching [22,23], MPM has emerged as a powerful modality for the imaging of tumors in many organs, including the prostate [24,25,26,27]. MPM generates tissue images by capturing several intrinsic contrasts. Contrast from two- and three-photon-excited intrinsic fluorescence from cellular biomolecules such as the metabolic cofactors nicotinamide adenine dinucleotide (NADH) and flavin adenine dinucleotide (FAD) enables imaging of cells [23,28,29,30]. Contrast from second-harmonic generation (SHG), a form of nonlinear scattering generated by noncentrosymmetric stromal collagen fibers (mostly collagen type 1 fibers), enables collagen-specific visualization of the ECM [31,32]. By simultaneously capturing multiple contrasts and spatially coregistering both cellular and extracellular features at subcellular resolution, label-free MPM is well suited to identifying features of the tumor microenvironment. Label-free MPM has been applied to profile the tumor microenvironment [33,34], and its advantage has been emphasized by reporting quantifiers for collagen, microvesicles, and metabolic changes not previously derived from H&E images [34]. MPM-identified collagen fiber orientation features have also been reported to be indicators of metastasis in PCa [17] and to have prognostic value in other human cancers [21,35]. However, although this compelling evidence suggests the utility of MPM in detecting features of aggressive tumors, detailed MPM characterization of the prostate tumors in the context of reactive stroma or poor clinical outcome status has not been previously performed.

In this study, we use MPM for high-resolution imaging of prostate tissue to identify collagen content, orientation, and fiber morphology signatures characteristic of reactive stroma and of tumors associated with a post-surgical clinical outcome (defined here by biochemical recurrence (BCR) following surgery). Images collected from unstained prostate tissue sections with well-annotated stromal regions were used to determine a set of MPM-identified features characteristic of reactive stroma. We subsequently imaged human PCa specimens to screen for differential collagen signatures between tumors from patients with BCR, and from patients who did not have recurrent disease. In our cohort we show an association between MPM-identified stromal collagen signatures of prostate pre-surgical or surgical tumor tissue and a post-surgical clinical outcome.

## 2. Materials and Methods

Patients and tissue samples: All human tissue investigations were approved by the Institutional Review Board of the Icahn School of Medicine at Mount Sinai (ISMMS), and informed consent was obtained from each subject included in this study.

Prostatectomy cohort: A tissue microarray (TMA) was constructed from 64 patients with PCa who underwent robotic-assisted radical prostatectomy (RP) by a single surgeon at the Mount Sinai Hospital between 2016 and 2017 without any previous form of adjuvant therapy. The TMA was prepared by sampling ~1 mm tissue cores from the formalin-fixed-paraffin-embedded (FFPE) prostatectomy block with the nodule with the higher Gleason score for each patient. For each patient, three tissue cores were sampled from the dominant tumor nodule. Each unstained tissue core was first imaged by MPM, followed by H&E labeling and imaging. We excluded patients for which the tissue cores had sectioning artifacts so that out of the 64 total patients, 59 were included in the MPM feature analyses. Baseline characteristics of the patients included in the analyzed radical prostatectomy (RP) cohort are summarized in Appendix A. Biochemical recurrence (two consecutive PSA levels >0.2 ng following surgery) within three years following surgery was seen in 11 (29.4%) patients.

Biopsy cohort: Imaging was also performed on unstained 5 μm FFPE sections from biopsy tissue cores from 17 patients, with nine of these patients developing biochemical recurrence within three years following surgery. All patients underwent MRI-targeted and systematic biopsies with subsequent robotic-assisted radical prostatectomy by a single surgeon at Mount Sinai Hospital between 2016 and 2017 without any previous form of adjuvant therapy. Baseline characteristics of these patients are summarized in Appendix A. For each patient, the biopsy core with the highest Gleason score was selected for imaging.

Image acquisition: All tissue sections were imaged with a commercial MPM microscope at the Microscopy CoRE and Advanced Bioimaging Center at ISMMS at room temperature and at 780 nm excitation. The emitted two-photon excited signal was detected by two photomultiplier tube (PMT) channels: (1) SHG at 390 nm (380–400 nm) of collagen in the backscattered mode, and (2) intrinsic fluorescence at 435–560 nm, showing general cellular and stromal morphology. Each channel is an 8-bit image with grayscale pixel intensity units ranging from 0 to 255. A high-resolution MPM tiled image (2D optical section) was obtained from each tissue section using a 25×/1.0 NA objective at 0.497 µm/pixel, at 4.0 µs/pixel, and at ~1.2 µm z-resolution. After MPM imaging, all sections underwent staining with H&E and were imaged with a Hamamatsu digital slide scanner at the Biorepository and Pathology CoRE at ISMMS and annotated by a pathologist (Dr. Brody).

Image processing: Selection of regions of interest: All MPM images were initially processed with Fiji version 2.3.1 (National Institutes of Health, Bethesda, MD, USA) to perform background subtraction. To allow comparisons between H&E and MPM images, each MPM image was registered to the corresponding H&E image of the imaged specimen by first using manual translation features in Fiji, followed by a 2D affine function in MATLAB version 2.3.1 (MathWorks, Natick, MA, USA). Areas with the dominant tumor were annotated on H&E images of the biopsy specimen by a pathologist, and these annotations were used to select two to three regions of interest (500 × 500 µm) on the corresponding MPM images. Regions of interest (ROI) were sampled throughout the annotated tumor area to provide a realistic representation for image analysis. For TMA images, the entire tissue core was considered a region of interest to be analyzed.

Image segmentation: We segmented each sample’s region of interest into glands and stroma surrounding the glands using the automatic boundary creation function of CurveAlign version V4.0 Beta (Laboratory for optical and computational instrumentation, University of Wisconsin, Madison, WI, USA) [36] software for fiber analysis of SHG images. This function segments glandular and stromal areas within an image based on inputs from collagen-specific SHG and the corresponding H&E image of the same site.

Collagen feature extraction: Fibrillar collagen area fraction and intensity quantification: MPM images were analyzed in ImageJ to determine the area fraction occupied by SHG-emitting collagen fibers in a region of interest (AF). The AF represents the percentage of pixels in the stroma occupied by collagen in the imaged tissue and is a measure of collagen amount or prevalence. For all MPM images, the SHG channel was thresholded using a threshold function in Fiji to separate the SHG signal from the background. A threshold was also set for the intrinsic fluorescence channel to determine the number of pixels occupied overall by tissue within the ROI. The AF was calculated by dividing the SHG pixels by the overall tissue pixels. The second collagen quantifier, SHG-emitting collagen fiber intensity (I_R_), is the mean pixel intensity value for all pixels in the SHG channel above the SHG threshold and is a measure of stromal collagen fiber brightness. Likewise, the green channel intensity (I_G_) is the mean pixel intensity value for all tissue pixels in the autofluorescence channel above the set threshold and is a measure of the overall stromal tissue brightness. To quantify the intensity of the SHG-emitting collagen signal in the red channel relative to the autofluorescence intensity in the green channel, we calculated a normalized stromal intensity ratio (I_R_/[I_R_ + I_G_]), where values closer to 1 indicate stromal composition dominated by bright SHG-emitting fibers.

Fibrillar collagen orientation and morphological features: The collagen fiber quantifiers’ width and length were extracted for each ROI image using the open-source software CT-FIRE version V2.0 Beta (Laboratory for optical and computational instrumentation, University of Wisconsin, Madison, WI, USA) [37]. The CT-FIRE algorithm allows for automated segmentation and extraction of individual collagen fibers from an image and for quantification of individual fibers by metrics including fiber length, fiber straightness, and fiber width. CT-FIRE produced histograms for each quantifier; we chose descriptive statistics such as the mean for each metric to quantify the fibers within each ROI. We also measured the bulk fiber alignment (coherence) and the localized fiber orientation (fiber angle) with respect to the tumor boundary by using the software CurveAlign V4.0 Beta [36]. The fiber alignment quantifier measures whether there is a preferred alignment of SHG-emitting fibers in the ROI, with values closer to 1 indicating a preferential alignment direction and values closer to 0 indicating isotropic distribution/no alignment. To determine the fiber angle, we used the automatic boundary creation module of CurveAlign to automatically segment the stromal-tumor gland boundaries based on coregistered SHG and H&E images. These morphological quantifiers and software are commonly used in cancer biology research to study collagen organization in different cancer types [38,39,40,41]. Feature extraction was performed using default parameters for CT-FIRE and CurveAlign. Appendix A summarizes the extracted collagen quantifiers. Appendix A summarizes the steps of the image analysis workflow to extract these quantifiers from each ROI.

Statistical analysis: For cohort description, all baseline clinical and pathological data such as age, PSA, race, biopsy Grade Group, radical prostatectomy Grade Group, extraprostatic extension, positive surgical margin, and lymph node invasion were reported as median and interquartile range for continuous variables or as frequencies and percentages for categorical variables. Continuous data were analyzed using unpaired two-sided Student’s *t*-tests for normally distributed data and Mann–Whitney U tests for nonparametric data. Similarly, chi-square/Fisher’s exact tests were performed on categorical data. Box plots were used to estimate MPM stromal content, fiber orientation, and morphological features in relation to reactive stromal status. Each boxplot point represents an MPM-derived quantifier per region of interest; the ends are the upper and lower quartiles, and the median is marked by a horizontal line inside the box. The whiskers extending from the boxes indicate variability outside the upper and lower quartiles. All tests were two-tailed with a *p*-value of 0.05 considered statistically significant. SAS version 9.4 software (SAS Institute Inc., Cary, NC, USA) was used. Cox proportional hazard regression and Kaplan–Meier survival analyses were performed to evaluate univariable associations between time to biochemical recurrence and the MPM-identified collagen features in the RP and biopsy cohorts. Univariable Cox associations were performed on continuous collagen variables. For the Kaplan–Meier survival analysis, continuous variables were divided into a “High” and “Low” group, with a cutoff value representing the mean of each continuous collagen variable. Multivariable Cox proportional hazards regression models were constructed to determine associations between time to biochemical recurrence and selected MPM-based collagen features when adjusted for clinical parameters. Only variables with a *p*-value < 0.05 at univariable analyses were included. Correlations between collagen variables were evaluated, and only collagen variables that were determined to be independent were considered in the same multivariable model. For example, the collagen area fraction, intensities, and fiber width variables were determined to be highly correlated (by Spearman correlation tests), and four separate multivariable models were built. Each model included all standard clinicopathological variables with a *p*-value < 0.05 at univariable Cox analyses. The AIC values and the likelihood ratio test were used to compare the separate multivariable models. Assessment of the proportional hazard assumption was performed to ensure no statistically significant violations of proportional hazards for all variables included in both the uni- and multivariable models. Analyses were performed in R package version 4.0.5 (The R Foundation for Statistical Computing, Vienna, Austria) and in SAS version 9.4 (SAS Institute Inc., Cary, NC, USA).

## 3. Results

### 3.1. Characterization of Normal and Reactive Stroma in Prostate Tissue Using MPM

To demonstrate the utility of MPM in detecting reactive stroma regions in prostate tumor tissue, we collected label-free, whole-slide, MPM mosaic images from 5 µm-thick FFPE tissue sections from a set of biopsy cores with known reactive stroma status. Each unstained tissue section was imaged by MPM, followed by H&E staining and annotation of reactive stroma regions by a pathologist for MPM visualization of normal and reactive prostate stroma (Figure 1). In the MPM images, intrinsic contrast is captured by two separate channels. The red channel detects SHG, a form of nonlinear scattering from anisotropic biological structures possessing large hyperpolarizabilities from fibrillar collagen (predominantly collagens I and II), and thus provides collagen-specific contrast. The green channel captures a broad two-photon excited intrinsic fluorescence (2PAF) signal that arises from both stromal and glandular structures. Thus, while the red channel represents collagen-specific contrast (arising from SHG-emitting collagen fibers), the green channel captures the overall tissue morphology. The overlay of these two channels allows for label-free visualization of stromal collagen relative to the tumor glands or the stromal-tumor boundary. Normal prostate stroma (Figure 1A–C) is composed mainly of smooth muscle cells, which have an overall light green appearance in the MPM images due to 2PAF from smooth muscle cells. Smooth muscle cells do not have collagen and do not emit SHG, and therefore have no red contrast. Extracellular-rich reactive stroma (RS) (Figure 1D–F) is characterized by loss of smooth muscle cells, expansion of the extracellular matrix, and collagen deposition. In the MPM images, extracellular-rich reactive stroma has a vivid red appearance due to SHG from the stromal collagen that now dominates the stromal composition instead of stromal smooth muscle cells. Therefore, compared with normal prostate stroma, reactive stroma has MPM-detectable characteristics; its vivid red appearance is associated with regions of extracellular-rich reactive stroma due to the increase in collagen fibers, whereas the light green is associated with regions of normal stroma due to the predominance of smooth muscle cells. Although collagen fibers are also visible in the H&E images, MPM provides enhanced collagen contrast and assessment of collagen-rich stromal regions within prostate tumors without exogenous labeling.

### 3.2. Quantifiable MPM-Identified Prostate Stromal Signatures

To convert the observed differences in stromal composition into quantifiable features, two to three regions with predominantly extracellular-rich reactive stroma (RS) or normal stroma (NS) were selected from each biopsy core, and a set of collagen content, orientation, and fiber morphology quantifiers were calculated for each region (Appendix A). We defined the stromal collagen content by three features: the area fraction of imaged tissue occupied by SHG-emitting collagen fibers (AF); the mean collagen fiber SHG (red channel) intensity (I_R_); and the normalized intensity, (I_R_/[I_R_ + I_G_]), where I_G_ represents the mean 2PAF (green channel) intensity value per imaged region. We used the collagen area fraction ratio (AF) to quantify the observed increase in the amount of SHG-emitting collagen fibers and the normalized intensity (I_R_/[I_R_ + I_G_]) to quantify the vivid intensity of the SHG-emitting collagen fibers in reactive compared to normal stromal regions. Consistent with visual observations, extracellular-rich RS has, on average, a statistically higher collagen area fraction (AF) (*p* < 0.001, by two-tailed Student’s *t*-test), as well as collagen fiber intensity IR (*p* < 0.001, by two-tailed Student’s *t*-test) and normalized stromal intensity (I_R_/(I_R_ + I_G_)) (*p* < 0.001, by two-tailed Student’s *t*-test) (Figure 1G) values than NS, indicating that its vivid red appearance is due to the increases in both the amount of SHG-emitting collagen as well as the collagen fiber SHG signal intensity. Next, we quantified stromal collagen orientation by the coherence for SHG-emitting fibers in each region, which indicates if fibers have a preferred alignment (coherence values closer to 1) or are randomly oriented (coherence values closer to 0). Collagen fiber morphology was quantified by the mean fiber width and fiber angle relative to the tumor edge in each imaged region. On average, compared with fibers in NS regions, RS regions have fibers that are more aligned (*p* < 0.01 by two-tailed Student’s *t*-test), thicker (*p* < 0.0001, by two-tailed Student’s *t*-test), and with a higher angle to the tumor gland (Figure 1G). These results support the utility of MPM for identifying microanatomic metrics of prostate stroma, enabled by the high-resolution and collagen-specific imaging.

### 3.3. MPM-Identified Prostate Stromal Features Associated with Biochemical Recurrence

Although MPM characterization of activated stroma regions is desirable, identification of stromal signatures associated with an important post-surgical clinical outcome, such as time to biochemical recurrence, indicates the potential of these MPM-identified features for identifying aggressive PCa. Therefore, we performed MPM imaging of unstained FFPE tissue cores from a tissue microarray from a cohort of 59 patients who underwent prostatectomy with 11 patients developing biochemical recurrence (BCR+) within a period of 3 years following surgery. Appendix A summarizes baseline clinical characteristics of the prostatectomy cohort. Tumor cores originating from BCR+ patients displayed a higher stromal SHG contrast (brighter red appearance) originating from the stroma surrounding the glands (Figure 2A,B) compared with the tumor cores originating from patients without recurrence (Figure 2C,D).

These observed differences were quantified by a set of stromal content, orientation, and fiber morphology signatures for each patient, based on first extracting signatures from each individual tissue core and then averaging the values of all tumor cores per patient. To establish if any of the MPM-identified signatures associate with time to biochemical recurrence, we first performed univariable Cox proportional hazards analyses (Table 1). At univariable Cox analyses, collagen area fraction, collagen fiber and normalized intensity, fiber width, and fiber angle to the tumor gland emerged to have a statistically significant association with biochemical recurrence (Table 1). Kaplan–Meier survival analyses also confirm that these variables are associated with recurrence-free survival. Specifically, high collagen content (collagen area fraction, fiber, or normalized stromal intensity) in prostate tumor prostatectomy tissue negatively impacts recurrence-free survival (Figure 2E). In addition, higher fiber width and higher fiber angle to the tumor glad also emerge to negatively impact recurrence-free survival. Next, multivariable analyses were performed to evaluate associations of the identified collagen variables with recurrence when adjusted for standard clinicopathological variables. Specifically, multivariable models were built that included collagen variables (Table 1) and clinical variables that emerged as statistically significant at univariable Cox analysis (Grade Group and stage at final pathology, and seminal vesicular invasion, Appendix A). The best performance was achieved by a multivariable model where collagen fiber SHG intensity signal and angle to the tumor gland are significantly associated with time to recurrence (Appendix A), further revealing the predictive potential of these MPM-identified signatures.

A fraction of the patients in the prostatectomy cohort had tumors with Grade Group of 2 or 3 at final pathology (*n* = 48), generally considered to be at intermediate risk of PCa recurrence. To demonstrate the association between MPM-identified stromal signatures and recurrence within this intermediate-risk group, we performed stromal feature characterization and univariable Cox analyses only on patients with Grade Group of 2 or 3 (11 BCR+ patients). At univariable analysis, collagen area fraction, collagen fiber and normalized stromal intensity, and fiber width emerged to be significantly associated with recurrence within this subgroup of patients (Appendix A), suggesting that these stromal signatures have the potential to refine risk assessment within this intermediate-risk group.

We subsequently imaged 5 μm-thick FFPE tissue sections from the dominant biopsy core from a small cohort of patients that had undergone prostatectomy with known BCR status for three years post surgery (BCR+ patients *n* = 9, BCR-patients *n* = 8). Consistent with results from the prostatectomy cohort, tumor sites in biopsy tissues from BCR+ patients (Figure 3A–C) display brighter SHG collagen contrast compared to BCR-patients (Figure 3D–F).

Based on H&E annotations for the dominant tumor site on each biopsy, 1–3 tumor regions were selected from each imaged biopsy; a set of MPM-identified stromal features was calculated for each region, and an average value per feature for each patient was calculated. At univariable analyses and consistent with prostatectomy tissue results, biopsy-derived collagen fiber and normalized intensity emerged to be significantly associated with time to recurrence (Table 2). Biopsy-extracted fiber morphology or orientation variables did not show any significant association with recurrence (Table 2). Kaplan–Meier survival analyses also show that patients with tumors characterized by high collagen fiber or normalized stromal intensity have shorter recurrence-free survival than patients with tumors with low collagen values (Figure 3G). Next multivariable analyses evaluate associations of biopsy-derived collagen fiber and normalized intensity and recurrence. (Appendix A). At multivariable analysis, only the collagen fiber intensity emerged to be significantly associated with time to recurrence (Appendix A).

## 4. Discussion

Distinguishing aggressive from indolent PCa presents oncologists, urologists, and pathologists with a clinical challenge. The present study examined the role of label-free microscopy profiling of the prostate tumor stroma to identify features associated with aggressive tumors and poor post-surgical outcomes. Our results demonstrate that reactive stroma, a predictor of post-surgical outcome in patients with PCa, can be characterized by MPM-identified signatures of collagen content, orientation, and morphology. Moreover, we showed that tumors from patients with poor post-surgical outcomes based on BCR+ status exhibit higher collagen fiber intensity signatures compared with tumors from nonrecurrent patients. This increase was detected in biopsy specimens as well as in prostatectomy tissues, suggesting the applicability of this MPM approach early in the management of patients with PCa. Identification of microscopic tumor changes using intrinsic sources of contrast also highlights the advantage of label-free high (subcellular)-resolution imaging approaches.

Our approach in determining image-based signatures of prostate tumor stroma was based on two key factors. The first factor is the application of label-free microscopy for collagen-specific subcellular resolution visualization of the stroma surrounding tumor glands in unstained tissue sections. MPM was used because of its ability to detect both collagen-specific as well as cellular contrasts [22,23], enabling interrogation of the stromal collagen fibers in relation to tumor glands. The second factor is image quantification by the extraction of collagen metrics as the basis for establishing stromal-based indicators of disease recurrence. The extraction of collagen quantity, orientation, and morphological quantifiers from MPM images of unstained tissue using semi-automated open-source computational tools has been previously used to derive new indicators for diagnosis and prognosis in other human tumor types [21,41]. Our study characterized reactive stroma surrounding prostate tumor glands by MPM-derived quantifiers of collagen content (fiber brightness and amount in tumor tissue), fiber orientation (coherence), and fiber morphology (fiber width, length, and angle to tumor boundary). Previously, the amount of reactive stroma in tumor tissue has been established as a survival predictor for PCa [14,15]; however, MPM-derived characteristics of reactive stroma have not been investigated as an alternative to H&E microscopy approaches. Herein we show that extracellular-rich RS regions have a distinct MPM-based appearance, quantified as a significant increase in collagen fiber amount and intensity. Moreover, collagen fibers in RS regions are also significantly more aligned (higher coherence value) and wider than in normal stroma, and features that have not previously been quantified from standard H&E examination. Although further MPM profiling of the different reactive stroma variants is needed [13], this new evidence reveals that reactive stroma is characterized by specific collagen organization and morphology signatures that are not routinely detected by standard histopathology evaluation.

ECM mechanical characteristics, such as matrix stiffness, regulate tumor and stromal cell behavior by mediating processes such as cell-matrix adhesion, proliferation, and migration [16]. During tumor progression, the ECM undergoes structural remodeling, such as increased deposition of collagens and proteoglycans, enhanced collagen crosslinking, as well as increased MMP-based collagen degradation [16]. The remodeled ECM is characterized by structural changes in interstitial collagens near tumor cells, which in turn affects gene expression, cell differentiation, and migration [16]. Collagen fiber features have been shown to have prognostic value in several human cancer types [21,41]. SHG-detected fiber coherence/alignment was previously found to be most useful in PCa research [17,42]. Previous studies in prostate biopsy tissues identified different fiber alignments in normal and malignant samples, with normal stromal areas having almost isotropic fiber alignment and high-grade tumor samples having a higher degree of alignment, suggesting the relevance of these metrics for diagnostic purposes [17,42,43]. However, there is limited evidence as to whether collagen or stromal features in prostate tissue are associated with poor post-surgical outcomes. In the present study, we used an MPM approach based on the characterization of prostatectomy specimens to show that collagen content and morphology characteristics of the stroma surrounding prostate tumor glands are associated with disease recurrence in a post-operational setting. We showed that high MPM-derived collagen area fraction, fiber SHG intensity signal, as well as high fiber width and angle with respect to the tumor gland from prostatectomy tumor tissues, were significantly associated with time to biochemical recurrence and negatively impacted recurrence-free survival at univariate Cox analyses. Most collagen prognostic studies to date have been performed on breast and pancreatic cancers and showed that increased collagen fiber alignment and fiber angles with respect to the tumor edge are prognostic for poor patient survival [20,44]. However, in line with our results, other studies have found that collagen fiber width has prognostic value, with increased collagen width predicting poor survival in patients with gastric and colon cancer [45,46].

We recognize that there are limitations associated with the present study. First, the reported MPM-based features were extracted from tissue regions of interest that were manually selected and covered a relatively larger tissue area (at least 500 × 500 μm), which may have introduced bias or in an inability to detect additional collagen features. Refining the criteria for selection of the quantified regions of interest may allow for improved detection of statistically significant fiber morphological or alignment differences between the examined patient outcome groups. In addition, the incorporation of machine learning techniques to automate region selection has the potential to address these limitations. Second, we have selected the examined tissues from convenient cohorts with an overrepresentation of the event and a relatively small sample size. Moreover, the small sample size allows more for a univariate than a multivariate analysis. Therefore, the results of our multivariate analyses must be interpreted with caution due to the limited sample size. Future studies will look at the clinical validity of our approach in a larger number of patients allowing for improved multivariate analysis and in longitudinal data sets. Third, this study was aimed at screening for stromal features associated with biochemical recurrence and not at developing a prediction model based on collagen signatures. Recent studies emphasize the importance of integrating multiple MPM-derived qualifiers into a single stromal-based signature to distinguish poor clinical outcomes rather than performing biomarker analysis on individual metrics [29,47,48]. We envision that our findings will serve as a foundation for further studies to develop methods for integrating multiple quantifiers into one stromal-based signature and for developing models for predicting PCa recurrence.

To the best of our knowledge, this is the first study to use MPM to identify stromal features in prostate tumors associated with a post-surgical clinical outcome. Our findings suggest that MPM-derived collagen features could provide information independent of traditional models for risk prediction and could potentially lead to new biomarkers to refine risk assessment of patients with PCa. Although this is a descriptive study, it nevertheless should provide momentum for new investigative efforts to establish the correlation of collagen signatures with other molecular and genomic signatures for PCa in an expanded cohort of tumor specimens for biologically relevant validation.

## Figures and Tables

**Figure 1 jpm-11-01061-f001:**
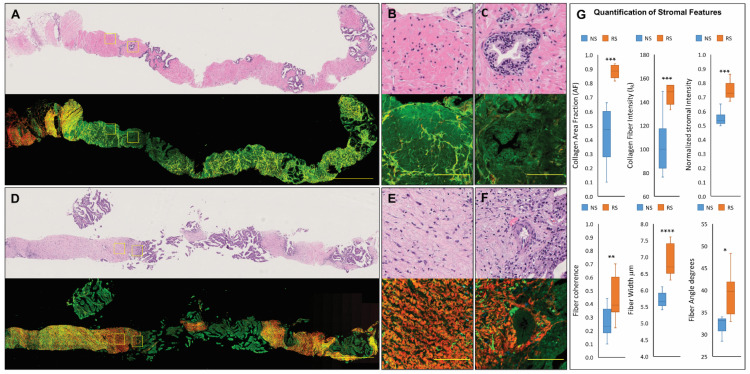
MPM of normal and reactive stroma in prostate tissue. H&E and MPM images of tumor-free prostate biopsy showing (**A**) overall morphology and zoomed-in views of (**B**) normal prostate stroma and (**C**) a tumor-free gland surrounded by normal stroma, and of a high-grade tumor biopsy showing (**D**) overall morphology and zoomed-in views of (**E**,**F**) reactive stroma surrounding a tumor gland. In the MPM images, normal stroma appears green due to two-photon excited autofluorescence from smooth muscle cells; reactive stroma appears red due to SHG emission for collagen fibers. Increased SHG contrast was visualized in all biopsies with extracellular-rich reactive stroma regions. Scale bars = 1000 µm in (**A**,**D**) and 100 µm in (**B**,**C**,**E**,**F**). (**G**) Box plots showing MPM stromal features in normal (NS) and reactive stroma (RS). Unpaired 2-tailed Student’s *t*-tests, all metrics were statistically significant, 10 imaged NS sites, 12 imaged RS sites (* *p* < 0.05, ** *p* < 0.01, *** *p* < 0.001, **** *p* < 0.0001). Stromal features: the fraction of tissue occupied by SHG-emitting collagen fibers (AF), collagen fiber intensity (I_R_), the normalized intensity I_R_/(I_R_ + I_G_), where I_G_ is the mean pixel autofluorescence intensity captured in the green channel, fiber orientation (degree of anisotropy/coherence), width, and angle with respect to tumor gland. The box plots show the median (central line), 25% (lower line), and 75% (upper line) quartiles for each group and the maximum and minimum values marked by the whiskers.

**Figure 2 jpm-11-01061-f002:**
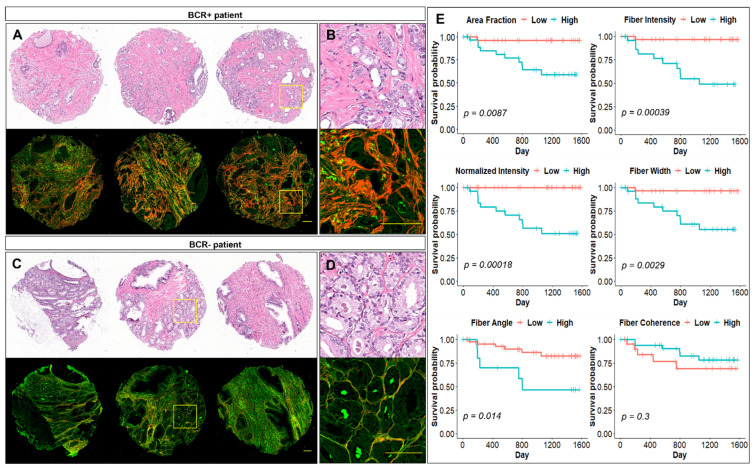
MPM-identified features from prostatectomy tissues associated with biochemical recurrence. H&E and corresponding MPM images of tumor cores from (**A**,**B**) a patient who developed biochemical recurrence (BCR+) and (**C**,**D**) a patient who did not develop biochemical recurrence (BCR−) following surgery. Scale bars = 100 µm. (**E**) Biochemical recurrence-free survival curves for collagen variables from the prostatectomy cohort. Continuous variables were first divided into a “Low” and a “High” group, at higher than the mean and lower than the mean of the variable being tested. Next, the Kaplan–Meier survival curves were determined for each variable. Blue lines indicate the “High” group, and red lines indicate the “Low” group. The log-rank *p*-values are shown in each plot.

**Figure 3 jpm-11-01061-f003:**
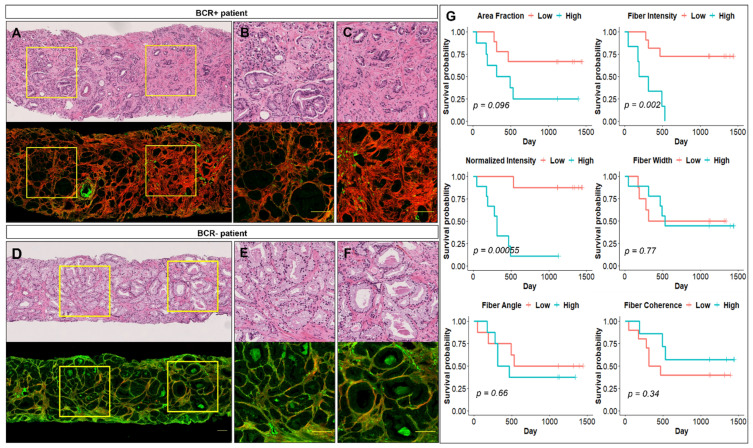
MPM-identified features from prostate biopsies associated with biochemical recurrence. H&E and corresponding MPM images of tumor areas from a biopsy core from (**A**–**C**) a patient who developed biochemical recurrence (BCR+) and (**D**–**F**) a patient who did not develop biochemical recurrence (BCR−) following surgery. Scale bars = 100 µm. (**G**) Biochemical recurrence-free survival curves for collagen variables from the biopsy cohort. Continuous variables were first divided into a “Low” and a “High” group, at higher than the mean and lower than the mean of the variable being tested. Next, the Kaplan–Meier survival curves were determined for each variable. Blue lines indicate the “High” group, and red lines indicate the “Low” group. The log-rank *p*-values are shown in each plot.

**Table 1 jpm-11-01061-t001:** Univariable Cox proportional hazards (PH) models to evaluate the association of time to biochemical recurrence and all collagen variables in the prostatectomy cohort.

Variable	Univariable Analysis			
	HR	95% CI		*p* Value >|z|
Collagen Area Fraction (AF)	1.05	1.01	1.09	0.014
Collagen Fiber Intensity (I_R_)	1.02	1.01	1.04	0.006
Normalized Intensity (I_R_/I_R_ + I_G_)	1.14	1.05	1.24	0.003
Collagen Fiber Length (mm)	1.07	0.95	1.21	0.245
Collagen Fiber Width (mm)	3.08	1.07	8.88	0.038
Collagen Fiber Angle (degrees)	1.25	1.01	1.55	0.041
Collagen Fiber Coherence	0.95	0.90	1.01	0.105

HR—hazard ratio, CI—confidence interval.

**Table 2 jpm-11-01061-t002:** Univariable Cox proportional hazards (PH) models to evaluate the association of time to biochemical recurrence and all collagen variables in the biopsy cohort.

Variable	Univariable Analysis			
	HR	95% CI		*p* Value >|z|
Collagen Area Fraction (AF)	1.05	0.99	1.11	0.082
Collagen Fiber Intensity (I_R_)	1.07	1.02	1.12	0.003
Normalized Intensity (I_R_/I_R_ + I_G_)	1.05	0.99	1.11	0.006
Collagen Fiber Length (mm)	0.93	0.68	1.28	0.669
Collagen Fiber Width (mm)	1.07	0.10	11.40	0.957
Collagen Fiber Angle (degrees)	1.08	0.88	1.33	0.463
Collagen Fiber Coherence	1.07	1.02	1.12	0.205

HR—hazard ratio, CI—confidence interval.

## Data Availability

Data sharing not applicable.

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
