# Peer review of "Multiphoton Microscopy for Identifying Collagen Signatures Associated with Biochemical Recurrence in Prostate Cancer Patients"

_jpm, 2021, doi:10.3390/jpm11111061_

Round 1

Reviewer 1 Report

The article, “Multiphoton microscopy (MPM) for identifying stromal signatures associated with biochemical recurrence in prostate cancer patients” by Pavlova et al, reports an association between MPM-identified stromal collagen features of prostate tumors and post-surgical disease recurrence. The authors performed MPM on prostatectomy specimens and identified collagen content, organization and morphological tumor signatures were extracted for each patient and screened for associated with recurrent disease. This manuscript is a commendable effort in trying to find a marker for screening of aggressive prostate cancer.  Here are the suggestions for manuscript improvement:

  1. The manuscript talks primarily about the collagen more than other stromal features; therefore, the title of the manuscript would be more appropriate if it mentions collagen features instead of more generalized stromal signature. Please change the title to better reflect the findings of the manuscript.

  1. There are other previously published studies based on MPM-identified collagen content and organization therefore for clarity it would be pertinent to remove or edit the sentences in the manuscript that says:

“Our study provides the first evidence for an association between MPM-identified…”.

“This is the first evidence to suggest an association between MPM-identified stromal collagen signatures…”

“Our study is the first to characterize reactive stroma surrounding prostate tumor glands by MPM-derived quantifiers of collagen content ….”

  1. A flow chart or table describing the steps of image analysis/ image processing could be presented in the manuscript to explain the work flow better.

  1. Figure 2 E is not readable, please replace with a higher quality image.

Author Response

We appreciate the effort of the reviewer in providing detailed and constructive comments to improve our manuscript. We responded accordingly as follows (for each individual point as listed below) and we believe these modifications have considerably improved the manuscript.

Response to Review 1:

  1. The manuscript talks primarily about the collagen more than other stromal features; therefore, the title of the manuscript would be more appropriate if it mentions collagen features instead of more generalized stromal signature. Please change the title to better reflect the findings of the manuscript.

Response: This is a valid point. In response, to better reflect the findings in the manuscript we changed stromal to collagen signatures in the title. Please see in revised manuscript the title is: “Multiphoton microscopy for identifying collagen signatures associated with biochemical recurrence in prostate cancer patients”

  1. There are other previously published studies based on MPM-identified collagen content and organization therefore for clarity it would be pertinent to remove or edit the sentences in the manuscript that says:

“Our study provides the first evidence for an association between MPM-identified…”.

“This is the first evidence to suggest an association between MPM-identified stromal collagen signatures…”

“Our study is the first to characterize reactive stroma surrounding prostate tumor glands by MPM-derived quantifiers of collagen content ….”

Response: We changes the sentences to:

Line 36: Our study shows an association between MPM-identified collagen features of prostate tumors and post-surgical disease recurrence, suggesting their potential for prostate cancer risk assessment.

Line 100: In our cohort we show an association between MPM-identified collagen signatures of prostate pre-surgical or surgical tumor tissue and a post-surgical clinical outcome.

Line 422: Our study characterized the reactive stroma surrounding prostate tumor glands by MPM-derived quantifiers of collagen content (fiber brightness and amount in tumor tissue), fiber orientation (coherence), and fiber morphology (fiber width, length, and angle to tumor boundary).

  1. A flow chart or table describing the steps of image analysis/ image processing could be presented in the manuscript to explain the workflow better.

Response: We included an additional supplementary table (Supplementary Table S3) to outline the steps of the image analysis workflow in the Supplementary Materials file. Original Supplementary Tables S3-7 are now labeled as Supplementary Tables S4-8, in the manuscript and in the supplement file.  Line 190 of the manuscript mentioned the addition of Supplementary Table S3.

  1. Figure 2 E is not readable, please replace with a higher quality image.

Response: We apologize for this issue. Figure 2E is updated so that the font and the line widths of all Kaplan-Meier plots is increased. This allows for more details to be visualized and makes the graphs readable. Figure 3G was also updated with the same graph edits.

Reviewer 2 Report

One of the goals in prostate cancer research is to find ways to discriminate between disease that will remain dormant for long periods in contrast to those cases when disease is aggressive with poor clinical outcomes. In this paper, Pavlova et al. use multiphoton microscopy (MPM) to analyze collagen content, organization, and morphological tumor signatures in a cohort of 59 patients that underwent prostatectomy and on biopsy specimens from 17 patients with known post-surgery recurrence status. The authors found that compared to tumors from patients whose disease did not recur, tumors from patients with recurrent disease exhibited higher MPM-identified collagen amount and collagen fiber intensity signal and width.

General comment:

The study is well designed, the results are clearly presented and the manuscript is well written. The authors fairly presented the weaknesses of the study.

Specific comments:

In an introductory or discussion part the authors could briefly mention the mechanism by which collagen content and organization influence the progression of the disease.

Lines 32-33: Something is wrong with the sentence. Maybe ‘’association’’ instead of ‘’associated’’?

Line 253: Period is missing at the end of the sentence.

Line 442: There seems to be an extra ‘’in’’.

Author Response

We appreciate the effort of the reviewer in providing constructive comments to improve our manuscript. We responded accordingly as follows (for each individual point as listed below) and we believe these modifications have considerably improved the manuscript.

Response to Review 2:

  1. In an introductory or discussion part the authors could briefly mention the mechanism by which collagen content and organization influence the progression of the disease.

Response: We added a brief mention in the discussion (line 437) regarding mechanisms of collagen remolding and its links to tumor and stromal cell behavior: “ECM mechanical characteristics, such as matrix stiffness, regulate tumor and stromal cell behavior by mediating processes such as cell-matrix adhesion, proliferation and migration [16]. During tumor progression the ECM undergoes structural remodeling, such as increased deposition of collagens and proteoglycans, enhanced collagen crosslinking, as well as increased MMP-based collagen degradation [16]. The remodeled ECM is characterized by structural changes in interstitial collagens near tumor cells, which in turn affects gene expression, cell differentiation and migration [16]. Collagen fiber features have been shown to have prognostic value in several human cancer types [21, 41]. SHG-detected fiber coherence/alignment was previously found to be most useful in PCa research [17,42]. “

  1. General comments:

Line 33: We edited ‘’associated’’ to “association”.

Line 254: We added the missing period.

Line 448: We removed the extra words.

Reviewer 3 Report

Report on “Multiphoton microscopy for identifying stromal signatures associated with biochemical recurrence in prostate cancer patients” by Ina P. Pavlova et al.

The manuscript analyzes the association between MPM-identified stromal collagen features of prostate tumors and post-surgical disease recurrence. The issue has a clear interest by its novelty in the field. I think the manuscript it is well written and quite clear. My concerns lie in some minor points:

  1. Although the authors clearly differentiate the univariate and multivariate analysis in their study, the minimum sample size allows more a univariate than a multivariate analysis. In the limitations of the study, it must be clearly stated that the results of the multivariate analysis have to be interpreted with caution due to the sample size.
  2. In the survival curves, provided in Figures 2 and 3, it is mandatory to provide the number of patients at risk in different time periods.
  3. In the legend of figure 1 authors informs that: “box plots show the median (central line) and 25% and 75% quartiles for each group”. This is incomplete, box plots show the median (central line) and 25% and 75% quartiles for each group and maximum and minimum values by whiskers in case of no outliers.
  4. Page 5, line 222, the reference of R is incomplete, you must specified the R version and the institution, for example R version 4.0.3 (The R Foundation for Statistical Computing, Vienna, Austria)

Author Response

We appreciate the effort of the reviewer in providing detailed and insightful comments to improve our manuscript. We responded accordingly as follows (for each individual point as listed below) and we believe these modifications have considerably improved the manuscript.

Response to Review 3:

  1. Although the authors clearly differentiate the univariate and multivariate analysis in their study, the minimum sample size allows more a univariate than a multivariate analysis. In the limitations of the study, it must be clearly stated that the results of the multivariate analysis have to be interpreted with caution due to the sample size.

Response: We modified the discussion to clearly state the limitations of the small cohort size for multivariate analyses. Line 473 now has the following addition: “Moreover, the small sample size allows more for a univariate than a multivariate analysis. Therefore, the results of our multivariate analyses must be interpreted with caution due to limited sample size. Future studies will look at the clinical validity of our approach in a larger number of patients allowing for improved multivariate analyses, and in longitudinal data sets.”

  1. In the survival curves, provided in Figures 2 and 3, it is mandatory to provide the number of patients at risk in different time periods.

Response: We appreciate the reviewer’s comment and agree that addition of the number of patients at risk in different time periods is important. However due to the small sample size we prefer to display the Kaplan Maier survival curves in Figure 2 and 3 in their current format. Our goal was to show that in our small cohort size there is preliminary evidence that the analyzed collagen metrics impact time to biochemical recurrence, rather than perform detailed analysis on patients at risk at different timepoint.

  1. In the legend of figure 1 authors informs that: “box plots show the median (central line) and 25% and 75% quartiles for each group”. This is incomplete, box plots show the median and 25% and 75% quartiles for each group and maximum and minimum values by whiskers in case of no outliers.

Response: We apologize for giving an incomplete description of the boxplot. The last sentence in Figure 1 is now modified to: “The box plots show the median (central line), 25% (lower line) and 75% (upper line) quartiles for each group and the with maximum and minimum values marked by the whiskers.

  1. Page 5, line 222, the reference of R is incomplete, you must specify the R version and the institution, for example R version 4.0.3 (The R Foundation for Statistical Computing, Vienna, Austria)

The reference in Line 221 is now modified to R package version 4.0.5 (The R Foundation for Statistical Computing, Vienna, Austria).